# MITF and TFEB cross-regulation in melanoma cells

**Josué Ballesteros-Álvarez**[1¤a], **Ramile Dilshat**[1], **Valerie Fock**[1¤b], **Katrín Möller**[1¤c], **Ludwig Karl**[1], **Lionel Larue**[2,3,4], **Margrét Helga Ögmundsdóttir**[5], **Eiríkur Steingrímsson**[1]*

**1** Department of Biochemistry and Molecular Biology, Faculty of Medicine, BioMedical Center, University of Iceland, Reykjavík, Iceland, **2** Institut Curie, PSL Research University, INSERM U1021, Normal and Pathological Development of Melanocytes, Orsay, France, **3** Univ Paris-Sud, Univ Paris-Saclay, CNRS UMR 3347, Orsay, France, **4** Equipe Labellisée Ligue Contre le Cancer, **5** Department of Anatomy, Faculty of Medicine, BioMedical Center, University of Iceland, Reykjavík, Iceland

¤a Current address: The Buck Institute for Research on Aging, Novato, CA, United States of America
¤b Current address: Department of Dermatology, Medical University of Vienna, Vienna, Austria
¤c Current address: Institute of Molecular Life Sciences, University of Zurich, Zurich, Switzerland
* eirikurs@hi.is

**Data Availability Statement:** The data is available on Figshare (DOI: 10.6084/m9.figshare.12568646).

## Abstract

The MITF, TFEB, TFE3 and TFEC (MiT-TFE) proteins belong to the basic helix-loop-helix family of leucine zipper transcription factors. MITF is crucial for melanocyte development and differentiation, and has been termed a lineage-specific oncogene in melanoma. The three related proteins MITF, TFEB and TFE3 have been shown to be involved in the biogenesis and function of lysosomes and autophagosomes, regulating cellular clearance pathways. Here we investigated the cross-regulatory relationship of MITF and TFEB in melanoma cells. Like *MITF*, the *TFEB* and *TFE3* genes are expressed in melanoma cells as well as in melanoma tumors, albeit at lower levels. We show that the MITF and TFEB proteins, but not TFE3, directly affect each other's mRNA and protein expression. In addition, the subcellular localization of MITF and TFEB is subject to regulation by the mTOR signaling pathway, which impacts their cross-regulatory relationship at the transcriptional level. Our work shows that the relationship between MITF and TFEB is multifaceted and that the cross-regulatory interactions of these factors need to be taken into account when considering pathways regulated by these proteins.

## Introduction

The microphthalmia-associated transcription factor (*MITF*) was discovered as the gene mutated in mice carrying the coat color mutation microphthalmia (*Mitf*) [1]. *Mitf* mutant mice lack melanocytes, resulting in pigmentation defects and deafness, and they have small eyes and some alleles show osteopetrosis (reviewed in [2]). In humans, *MITF* mutations have been linked to the rare dominant pigmentation disorders Waardenburg Syndrome type 2A (WS2A) [3,4] and Tietz Syndrome (TS) [5] as well as the more serious COMMAD syndrome in compound heterozygotes [6]. In addition, the *MITF* germline mutation E318K, has been

**Funding:** ES, 130230-052, Research fund of Iceland, www.rannis.is ES, 163413-051, Research fund of Iceland, www.rannis.is.

**Competing interests:** The authors declare no competing interests.

linked to melanoma [7,8]. MITF is regarded as the master regulator of the melanocyte lineage as it regulates the expression of various genes required for melanocyte development, proliferation and survival (reviewed in [2]). The *MITF* gene is expressed in multiple isoforms that differ in their first exon and promoter usage [2,9]. In most isoforms, the variable first exon is spliced to exon 1B1b which is then spliced to exon 2 and the following common exons which encode for the functionally important motifs necessary for DNA-binding, protein dimerization and transactivation ability [2,9]. Among the exon 1B1b-containing isoforms, MITF-D is expressed in the human retinal pigment epithelium (RPE) [10], whereas isoforms MITF-A, MITF-B, MITF-E and MITF-H are more ubiquitous [9,11]. The shortest isoform, termed MITF-M, is predominant in melanocytes and contains a short exon 1M directly spliced to exon 2 [12].

Together with transcription factor EB (TFEB), TFE3 and TFEC, MITF forms a subfamily of related bHLHZip proteins, sometimes termed the MiT-TFE family [1,13,14]. TFEB and TFE3 participate in the biogenesis of lysosomes and autophagosomes and the clearance of cellular debris upon starvation or lysosomal stress, through the activation of the CLEAR (Coordinated Lysosomal Expression and Regulation) network of target genes [15–20]. More recently, a role has been described for MITF in regulating the starvation-induced autophagy response [21]. The bHLHZip transcription factors, including the MiT-TFE subfamily, form homo- and/or heterodimers that bind to DNA and activate target genes. *In vitro* translated MITF has been shown to be able to form stable DNA-binding heterodimers with TFEB, TFE3, and TFEC [14]. These proteins have nearly identical basic regions and very similar HLH and Zip domains [1]. However, they fail to dimerize with other related bHLHZip factors such as c-Myc, MAX or USF [14] due the presence of a 3 amino acid sequence in the MiT-TFE proteins, which limits dimerization within that family [22,23].

Interestingly, the MITF-M isoform, which is predominantly expressed in melanocytes, is constitutively nuclear [24,25]. This is in stark contrast to TFEB, TFE3 and other MITF isoforms, which have been shown to be located in the cytoplasm under normal conditions [26,27]. The predominant nuclear localization of MITF-M has been explained by the absence of an N-terminal domain important for cytoplasmic retention, encoded by exon 1B1b [26,28]. This 30 amino acid cytoplasmic retention domain is present in TFEB, TFE3 and all isoforms of MITF except for MITF-M [15,28]. This domain allows for interactions with active Rag-GTPase heterodimers, which are required for the localization of these factors to the lysosome [28]. In the case of TFEB, phosphorylation of TFEB by kinases such as mTORC1 [29] and ERK2 [19] promotes interaction with 14-3-3 proteins leading to subsequent cytoplasmic retention of TFEB [26].

In this study, we show that the MITF, TFEB and TFE3 transcription factors are all produced in melanoma cells, albeit at different levels. MITF and TFEB can regulate each other's expression, and the mTOR signaling pathway further regulates this cross-regulatory relationship by modulating their subcellular localization and transcriptional activity. Thus, pharmacological interventions that modulate the expression or activity of these factors may affect the balance in their expression and the cellular processes that they regulate.

## Results

### MITF and TFEB modulate each other's expression

Previous analysis of RNA sequencing data from 368 metastatic melanoma tumors from The Cancer Genome Atlas (TCGA) [30] showed that the mRNA expression of *TFE3*, *TFEB*, and *TFEC* was 4-, 14-, and 40-fold lower than that of *MITF*, respectively [21]. Furthermore, analysis of gene expression in 23 human melanoma cell lines as well as in normal human epidermal melanocytes (NHEM) using a microarray platform revealed that the expression of *TFEB* and

*TFE3* was roughly 50-fold lower than that of *MITF*, whereas expression of *TFEC* mRNA was about 850-fold lower than that of *MITF* [21,31]. Particularly, in the 501Mel and Skmel28 human melanoma cell lines used in this study, *MITF* is highly expressed whereas *TFEB* and *TFE3* are expressed at considerably lower levels. *TFEC* is virtually undetectable in both cell lines (S1 Table). Although the expression level of transcription factors may not be directly related to their importance, we decided to focus on MITF, TFEB and TFE3 in the remaining analysis.

The finding that the MiT-TFE factors are co-expressed in melanoma cells and tumors, together with recent evidence pointing to a role of MITF in the regulation of lysosomal and autophagy-related genes [21,32], similar to TFEB and TFE3, may suggest an overlap in function or cooperation between these factors. This led us to investigate whether MITF and TFEB could transcriptionally regulate each other's expression. Publicly available ChIP-seq data for MITF in 501Mel [33] and Colo829 human melanoma cells [34] were analyzed in order to determine if MITF might be involved in directly regulating its own expression or that of TFEB and TFE3. A number of ChIP-seq peaks containing CACGTG or CATGTG elements were observed for MITF in the *MITF* gene, in both 501Mel and Colo829 cells (Fig 1A). In addition, both ChIP-seq datasets showed that MITF binds to a region within intron 1 of *TFEB* containing an E-box CAGCTG sequence (Fig 1B). In contrast, no ChIP-seq peaks were found within or near the *TFE3* gene (Fig 1C). The peak closest to the *TFE3* gene is an E-box element located approximately 35 kb upstream of the transcription start site (TSS), between the neighboring genes *WDR45* and *PRAF2*, and was not considered further (10.6084/m9.figshare.12568646). These data suggest that MITF can regulate the expression of both *MITF* and *TFEB* through binding to putative DNA regulatory elements, whereas MITF is less likely to directly regulate the expression of *TFE3 in vitro*.

In order to determine whether MITF and TFEB are able to influence each other's expression, we investigated the effects of transiently overexpressing the two individual factors on their expression in 501Mel and Skmel28 human melanoma cell lines. We first separately over-expressed MITF or TFEB in 501Mel cells containing the doxycycline (DOX)-inducible piggy-bac (pBac) vectors pBac-MITF or pBac-TFEB prior to evaluating the mRNA levels of each factor by qRT-PCR. The overexpressed MITF protein was the (-) isoform lacking exon 6a that encodes six amino acids located immediately upstream of the basic domain [12]. We used primers specific for the MITF(+) isoform to determine the expression of endogenous *MITF* mRNA and universal MITF(+/-) primers to detect expression of total *MITF* mRNA; *TFEB* and *TFE3* were assayed using gene-specific primers (S2 Table). Overexpressing MITF(-) significantly increased *TFEB* mRNA expression whereas *TFE3* levels remained unchanged (Fig 2A). In addition, MITF(-) overexpression significantly reduced the expression of endogenous *MITF* as detected using primers specific to the *MITF(+)* isoform (Fig 2A). Overexpression of TFEB in the 501Mel cell line resulted in reduced *MITF* expression whereas *TFE3* mRNA levels were unaffected (Fig 2B). These results were confirmed at the protein level by Western blot analysis—overexpression of TFEB resulted in reduced MITF protein expression (Fig 2C). Ectopic expression of MITF led to decreased levels of endogenous MITF (Fig 2C) and increased TFEB protein levels (Fig 2D). Quantification of the changes in protein expression were performed by normalizing each protein's band intensity to the expression of Actin and are presented as a fold-change relative to the samples overexpressing an empty vector (Fig 2E).

To further examine the effect of MITF on TFEB expression, we treated 501Mel cells overexpressing FLAG-tagged MITF in a DOX-inducible pBac vector with increasing doxycycline concentrations and assayed for TFEB expression using Western blot analysis. As expected, we observed a dose-dependent increase in MITF-FLAG protein expression after 24 hours of doxycycline treatment (Fig 2F and 2G). Importantly, increasing expression of the MITF protein led

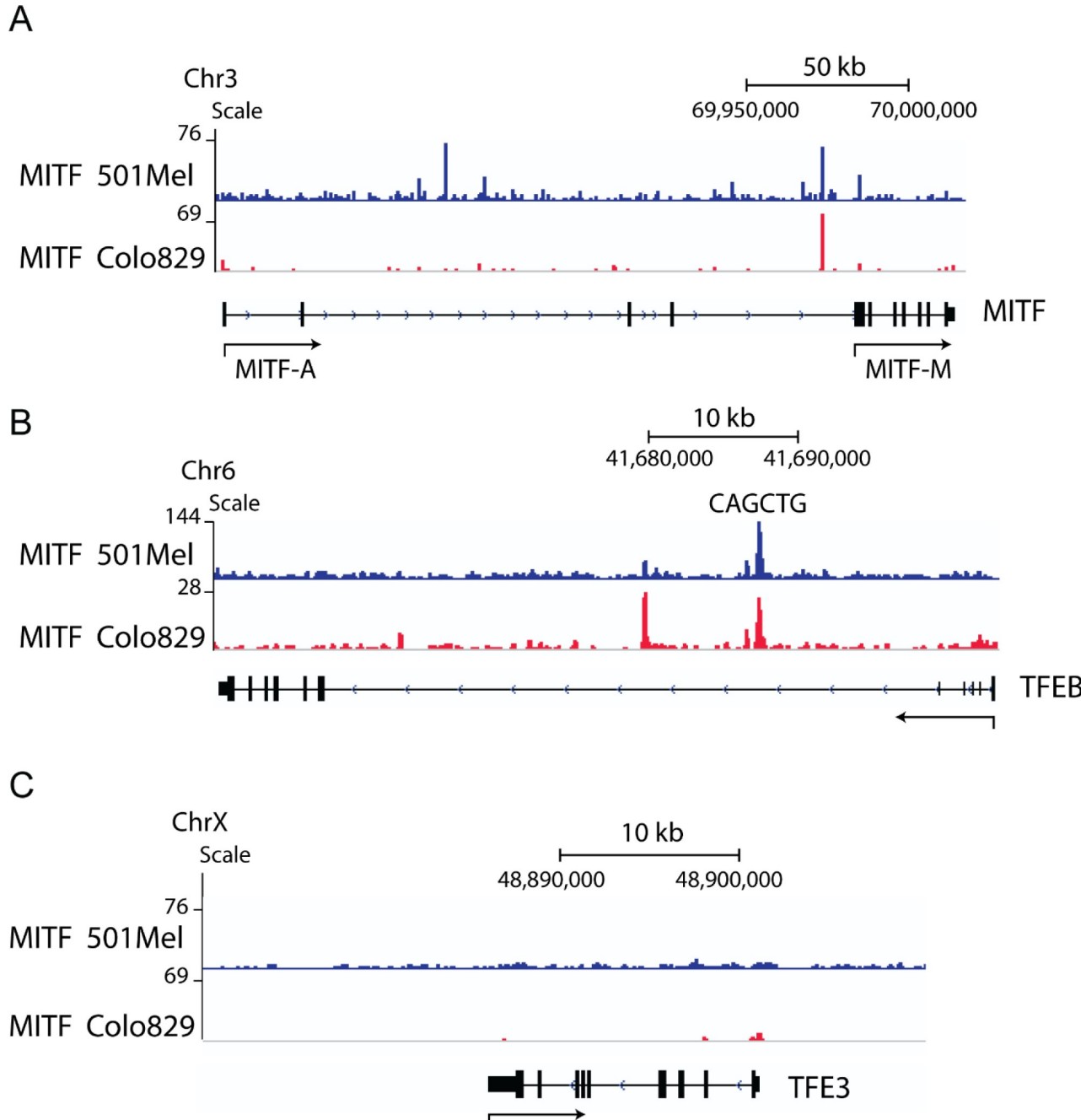

**Fig 1. MITF binds to the *MITF* and *TFEB* genes in melanoma cells.** MITF ChIP-seq data in 501Mel and Colo829 melanoma cells show peaks for MITF in the *MITF* gene (A) and a peak in the intron 1 of *TFEB* containing a CAGCTG regulatory element (B). No binding sites were observed within the *TFE3* gene (C) nor within 20 kb upstream or downstream (10.6084/m9.figshare.12568646).

to a significant increase in TFEB protein expression, thus supporting the positive effects of MITF on TFEB (Fig 2F and 2G). Since the MITF-FLAG pBac system overexpressed the MITF (+) isoform, it is unlikely that there are differences between the MITF (+) and (-) isoforms with respect to effects on TFEB expression. In order to investigate this further, we transiently overexpressed the MITF(+) isoform in the 501Mel cell line and analyzed mRNA expression of

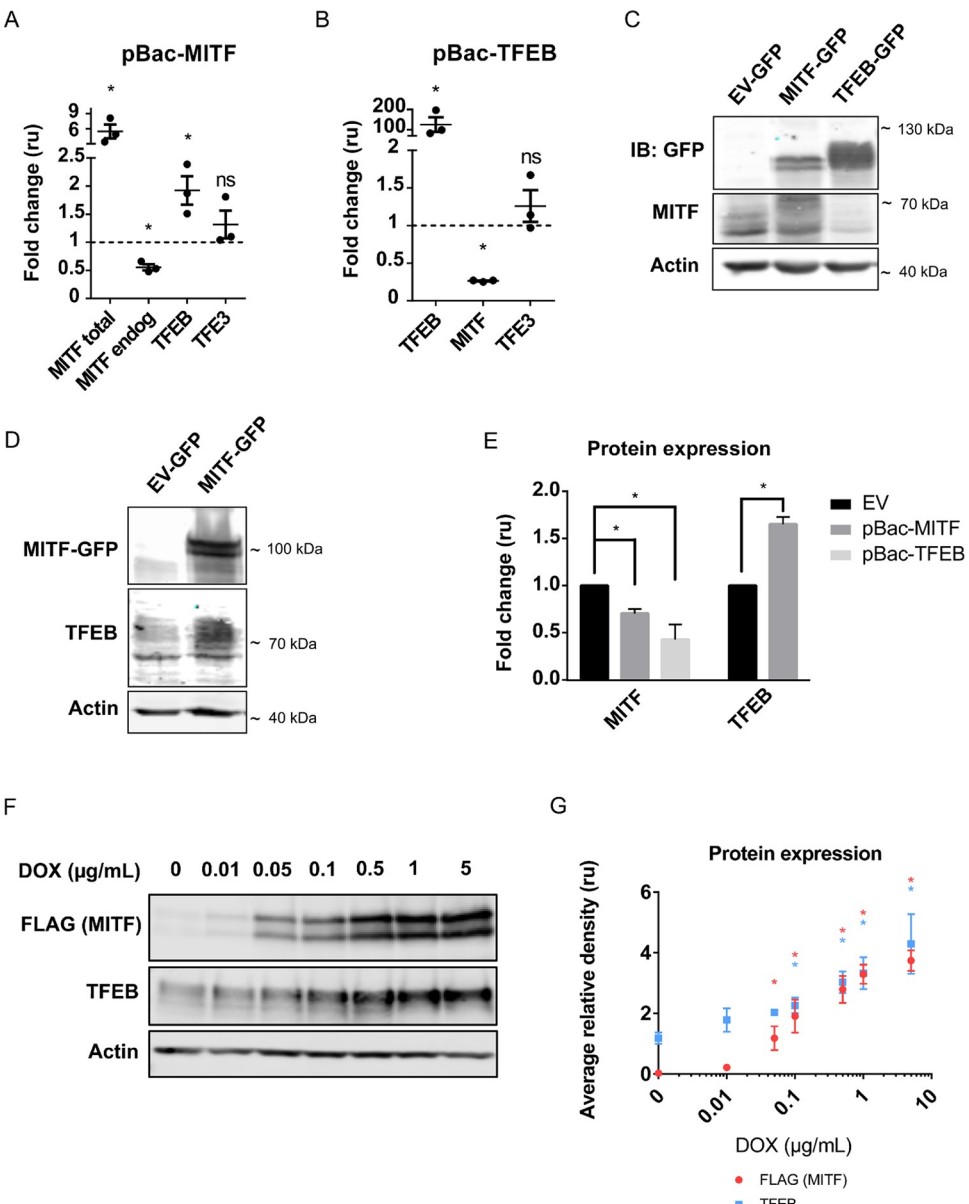

**Fig 2. MITF and TFEB modulate each other's expression upon overexpression in 501Mel cells.** (A-B) The expression of *MITF, TFEB* and *TFE3* as determined by RT-qPCR after overexpression of MITF(-) (A) or TFEB (B) in 501mel cells compared to empty vector (EV). Bars represent SEM. * indicates significance at p<0.05. (C-D) Western blot analysis of MITF (C) and TFEB (D) proteins, using specific antibodies, after overexpression of TFEB (C) and MITF (C, D) in 501mel cells. Shown is a representative figure for at least three independent experiments. (E) Quantification of the Western blots presented in C-D. Bars represent SEM. * indicates significance at p<0.05. (F) Western blot analysis of MITF and TFEB proteins, using FLAG (MITF) or TFEB-specific antibodies, upon dose-dependent doxycycline-inducible overexpression of FLAG-tagged MITF in 501mel cells. Shown is a representative figure for three independent experiments. (G) Quantification of the Western blot presented in F. Bars represent SEM. * indicates significance at p<0.05.

*TFEB* and endogenous *MITF* using 3'UTR-specific primers. The effects were comparable to those observed after overexpression of the (-) isoform, further validating that these six amino acids do not have additional effects on the expression of MITF or TFEB (S1A Fig). We also

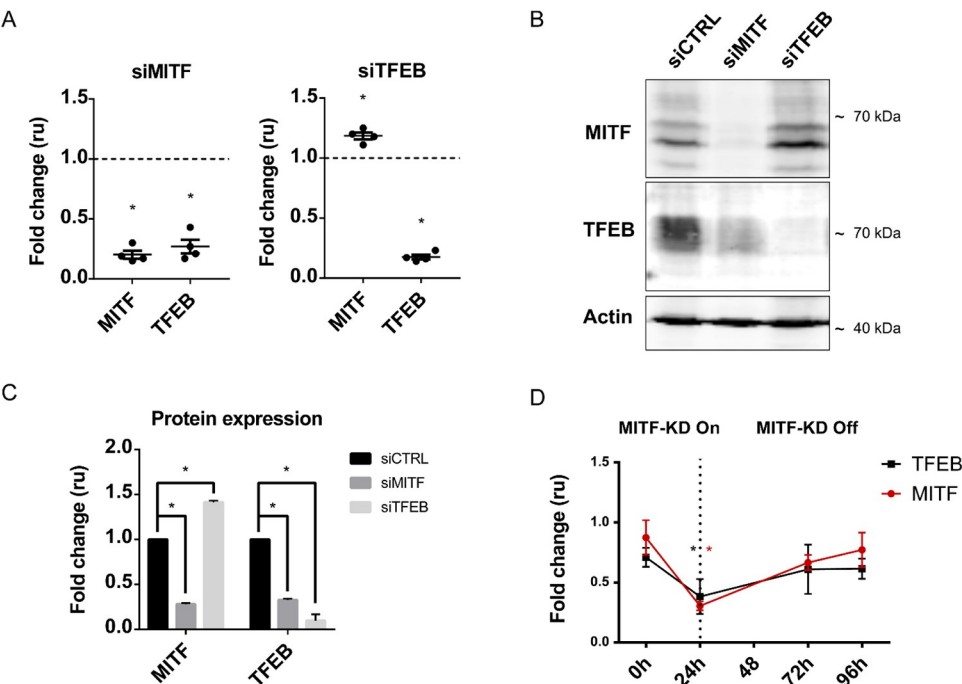

**Fig 3. MITF and TFEB modulate each other's expression upon knockdown in 501Mel and Skmel28 cells.** (A) The expression of *MITF* and *TFEB* as determined by RT-qPCR after siRNA-mediated knockdown of each factor compared to control siRNA in 501Mel cells. Bars represent SEM. * indicates significance at p<0.05. (B) Western blot analysis of MITF and TFEB proteins after siRNA-mediated knockdown of MITF or TFEB in 501 Mel cells. Shown is a representative figure and quantification of three independent experiments (C). Bars represent SEM. * indicates significance at p<0.05. (D) The expression of *MITF* and *TFEB* as determined by RT-qPCR after doxycycline-induced (1μg/ml) miR-MITF knockdown, compared to miR-NTC control in Skmel28 cells; three independent experiments are shown. Bars represent SEM. * indicates significance at p<0.05.

performed overexpression of MITF(+) and TFEB in Skmel28 cells and observed similar effects as observed in the 501Mel cells (S1B Fig).

In order to validate our overexpression experiments, we used two sets of pooled siRNAs to separately knock down MITF or TFEB in 501Mel cells followed by RT-qPCR and Western blot analysis. The smart pool of siRNAs utilized were effective at targeting each individual factor as confirmed at both mRNA and protein levels (Fig 3). We observed that MITF knockdown dramatically reduced expression of both *TFEB* mRNA (Fig 3A) and protein (Fig 3B) to a degree comparable to that observed upon TFEB knockdown. On the other hand, TFEB knockdown increased expression of the MITF mRNA (Fig 3A) and protein (Fig 3B). Quantification of the changes in protein expression were performed as the mean of three independent experiments by normalizing each protein's band intensity to the expression of Actin and are presented as a fold-change relative to the samples treated with control siRNA (Fig 3C). Of note, TFEB protein could not be detected with our TFEB-specific antibody in Skmel28 cells. However, in Skmel28 cells, siRNA-mediated silencing of MITF reduced the mRNA expression of *TFEB* (S2 Fig). To further characterize the effects of siRNA-mediated silencing of MITF on *TFEB* expression, we employed an Skmel28 cell line expressing a doxycycline-inducible miRNA targeting MITF. The miR-MITF-mediated knockdown of MITF correlated with a significant reduction in *TFEB* mRNA levels 24 hours after addition of doxycycline into the cell culture medium (Fig 3D). Importantly, after 24 hours of MITF knockdown, we washed off doxycycline treatment, and observed that *TFEB* mRNA was gradually restored at 72 and 96

hour doxycycline-free time points along with *MITF* mRNA ([Fig 3D]). Collectively, these data indicate that MITF and TFEB are able to regulate each other's mRNA and protein expression in human 501Mel and Skmel28 melanoma cells.

## MITF directly regulates *TFEB* expression

We then asked whether the effects of MITF on the expression of *TFEB* are direct transcriptional effects involving DNA-binding by MITF. To address this, we cloned the DNA sequences in *TFEB* shown to be bound by MITF according to ChIP-seq analysis ([Fig 1B]), into a pGL3-Promoter luciferase reporter plasmid. More specifically, we chose a fragment encompassing an 853-basepair (bp) sequence of intron 1 of *TFEB* containing the region between bases -29,373 and -28,521 located upstream of exon 2 of *TFEB*. This sequence contains a CAGCTG sequence, a potential MITF binding site ([Fig 4A]). A mutated version of this construct was generated where the potential binding element was mutated to CCCTTT. The resulting reporter constructs were transfected into HEK293T cells, which express low levels of the MiT/TFE transcription factors endogenously, together with a construct expressing a FLAG-tagged wild type MITF-M protein. We used the *Tyrosinase* (*TYR*) promoter as a positive control, a classical MITF target [35]. The results revealed transactivation of the *TYR* promoter by MITF, showing that our assay worked as intended ([Fig 4B]). Expression from the wild type *TFEB* intron 1 element was significantly increased upon expression of FLAG-tagged MITF-M. However, this enhanced transactivation was abrogated when the 6-bp sequence was mutated ([Fig 4B]), suggesting that MITF binds to this sequence and activates expression of *TFEB*. We further tested whether the effects on *TFEB* expression upon ectopic expression of

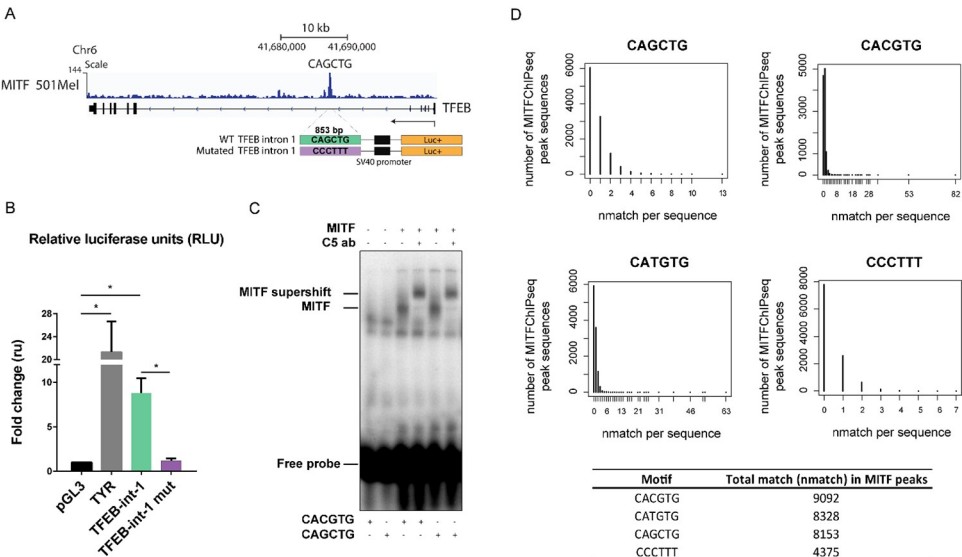

**Fig 4. MITF activates TFEB expression through a CAGCTG motif in intron 1.** (A) Schematic representation of the luciferase reporter constructs generated for the MITF binding sites observed in the *TFEB* gene. (B) HEK293T cells were transiently co-transfected with a p3XFLAG-CMV-14 construct with or without MITF-M (empty vector, EV) and reporter constructs and assayed for luciferase activity after 24h. Luminescence signal is expressed as fold change over an empty luciferase reporter for *Tyrosinase* and for a luciferase reporter containing the fragment from *TFEB* intron 1 and a mutated version thereof in front of the minimal SV40 promoter. Error bars represent the SEM of three experiments. * indicates significance at p<0.05. (C) EMSA showing binding of *in vitro* translated MITF protein to oligos containing the CACGTG and CAGCTG sequences. Supershifts with the C5 MITF specific antibody are indicated, confirming the specificity of the gel shifts. (D) A total of 11,173 sequences under the MITF ChIP-seq peaks were called with p<0.05. Graphs show the number of peaks among all the peaks called with any given number of matches for each motif. Bottom table shows the total number of matches for each motif in all the peaks.

MITF involved direct transcriptional regulation by using a mutant MITF protein, which lacks four arginines in the DNA binding domain that are essential for both DNA binding and nuclear localization of MITF. R214-217A mutant MITF exhibits constitutive cytoplasmic localization and presumably is transcriptionally inactive [25]. Overexpression of this R214-217A MITF in HEK293T cells showed no reporter transactivation from the *TYR* promoter or the wild type *TFEB* intron 1 element (S3 Fig), suggesting that the reporter transactivation previously observed requires the presence of transcriptionally active MITF protein.

Interestingly, the MITF binding site in intron 1 of *TFEB* is a CAGCTG element and not a canonical CACGTG E-box or a CATGTG M-box, which have previously been described as target sequences for MITF transcriptional regulation [36,37]. In order to validate that MITF can efficiently bind the CAGCTG sequence, we used electrophoretic mobility shift assay (EMSA) to determine whether an *in vitro* translated MITF-M protein can effectively bind a radiolabeled probe containing the above sequence. In this assay, we included a radiolabeled E-box (CACGTG) as a positive control for MITF-DNA binding. As expected, the MITF protein can shift the canonical E-box (Fig 4C). It can also shift the CAGCTG probe with similar efficiency (Fig 4C). The DNA-protein complexes were both further shifted with the anti-MITF C5 antibody, indicating that the complexes observed specifically bind MITF (Fig 4C). The CAGCTG 6-mer is not a canonical E-box or M-box, both of which have been previously described as MITF binding sites. We thus analyzed the MITF ChIP-seq dataset ($p < 0.05$) [33] in order to find whether there are more occurrences of the CAGCTG regulatory element among MITF target genes. In the ChIP-seq dataset, 11,173 sequences under the peaks were found to contain putative MITF binding sites. The CACGTG motif was present at least once in 9,092 of the 11,173 sequences. The CATGTG element, core to the M-box regulatory element associated with several melanocyte-specific genes [38,39], was present in 8,328 sequences. The CAGCTG motif (the motif bound by MITF in the TFEB promoter) was found in 8,153 sequences. In contrast, the CCCTTT motif used as a scramble control was present in only 4,375 peaks (Fig 4D). These results indicate that MITF directly regulates its target genes through direct binding to CANNTG motifs, including the transcription of *TFEB* through a CAGCTG motif located in *TFEB* intron 1.

## mTOR signaling affects the subcellular localization of MITF and TFEB in melanoma cells

The subcellular localization of TFEB and TFE3 as well as of the MITF-A isoform has been shown to be regulated by the mTOR pathway [40]. Phosphorylation of TFEB at Ser211 by mTORC1 promotes its cytoplasmic localization [26]. In contrast, MITF-M, the main isoform of MITF in melanocytes and melanoma cells has been shown to be primarily nuclear [25]. We analyzed the endogenous expression of MITF and TFEB in the 501Mel and Skmel28 cell lines by immunostaining and confocal microscopy. In addition, we used the mTOR pan-inhibitor Torin-1 to determine whether their subcellular localization responds to inhibition of this signaling pathway. Both endogenous MITF and TFEB were detected in 501Mel cells using specific antibodies (Fig 5A). While TFEB was located in the cytoplasm and nucleus of 501Mel cells (Fig 5A), MITF showed a major nuclear presence although a fraction of the protein was present in the cytoplasm (Fig 5A). Treating 501Mel cells with the mTOR pan-inhibitor Torin-1 (1 μM, 3 hours) resulted in increased nuclear localization of the endogenous TFEB protein, suggesting that mTOR activity contributes to the cytoplasmic retention of TFEB in melanoma cells (Fig 5A). Interestingly, the mTOR inhibitor also increased the fraction of MITF located in the nucleus (Fig 5A). The same results were observed when constructs containing GFP-fusions of the MITF-M and TFEB proteins were overexpressed in 501Mel cells using a doxycycline-

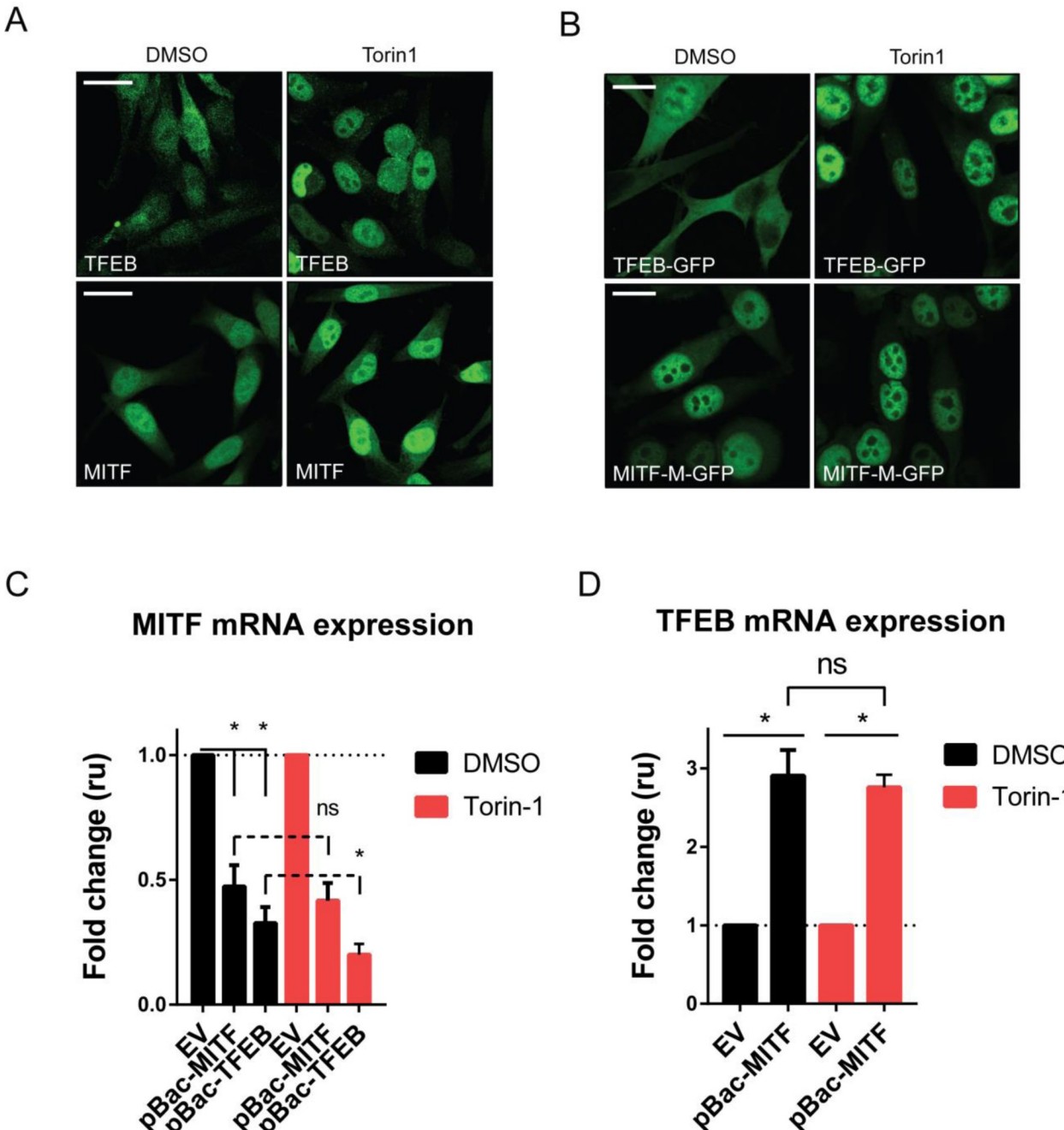

**Fig 5. mTOR signaling affects the subcellular localization and cross-regulation of MITF and TFEB.** (A) Immunofluorescence images of human 501Mel cells after treatment with vehicle (DMSO) or an mTOR inhibitor (Torin-1, 1 μM, 3 hours) showing endogenous TFEB and MITF proteins in green. (B) Human 501Mel cells expressing doxycycline-inducible GFP-tagged MITF-M or TFEB were treated with vehicle (DMSO) or Torin-1, then fixed for GFP imaging. (C) The expression of *MITF* as determined by RT-qPCR upon overexpression of MITF(-) or TFEB with or without mTOR inhibition for 3 hours (Torin-1, 1 μM). (D) The expression of *TFEB* as determined by RT-qPCR upon overexpression of MITF(-) with or without mTOR inhibition (Torin-1, 1 μM, 3 hours) compared to vehicle (DMSO). Error bars represent SEM. * indicates significance at p<0.05. "ns" indicates no significance at p>0.05.

inducible pBac system, and subsequent treatment with Torin-1. The GFP-fusion of TFEB was mostly cytoplasmic whereas that of MITF was mostly nuclear, suggesting that overexpression

does not affect the nucleocytoplasmic distribution of these factors (Fig 5B). Torin-1 treatment led to nuclear localization of TFEB and a reduction in the cytoplasmic presence of MITF. This shows that the cytoplasmic portion of the MITF-M isoform is affected by mTOR signaling and that the cytoplasmic signal detected with an antibody against endogenous MITF is neither an artifact nor the result of cross-reactivity and is not due to the presence of alternative isoforms of MITF in these cells. Immunostaining of Skmel28 cells showed that MITF is expressed in these cells whereas TFEB protein was not detectable (S4 Fig). Similar to 501Mel cells, MITF was mostly nuclear in this cell line and its cytoplasmic presence was reduced after treatment with the mTOR inhibitor (S4 Fig).

Next we hypothesized that the mTOR-induced changes in the subcellular localization of MITF and TFEB may affect their respective transcriptional regulation. We have shown that both MITF and TFEB negatively affect the expression of endogenous MITF in both melanoma cell lines used in this study (Figs 2, 3 and S1 Fig) and would therefore expect their increased nuclear presence upon Torin-1 treatment to further repress MITF expression. 501Mel cells containing the doxycycline-inducible pBac-MITF or pBac-TFEB vectors were treated with Torin-1 or a vehicle control (DMSO) prior to evaluating the mRNA levels of each factor by qRT-PCR. Overexpression of either MITF or TFEB inhibited the expression of endogenous *MITF* mRNA in 501Mel cells (Fig 5C), as previously shown (Fig 2). The expression of endogenous *MITF* in pBac-MITF overexpressing cells was not further reduced upon Torin-1 treatment (Fig 5C), however it was further decreased in Torin-1 treated cells containing pBac-TFEB as compared to vehicle-treated cells (Fig 5C). These results fit the fact that MITF is consistently nuclear whereas TFEB is both cytoplasmic and nuclear but becomes predominantly nuclear upon treatment with Torin-1 and therefore can further reduce *MITF* gene expression. As expected, the pBac-MITF cells showed increased *TFEB* mRNA expression; this induction was not further enhanced by Torin-1 treatment (Fig 5D). Taken together, these data indicate that the mTOR signaling pathway is active in melanoma cells and can affect import of TFEB into the nucleus. Increased nuclear localization of TFEB may subsequently enhance its transcriptional repression of *MITF*. In contrast, the melanocyte and melanoma cell-specific MITF-M isoform is mostly nuclear under basal conditions and highly expressed in several melanoma cell lines and tumors. Therefore, the subtle mTOR-dependent effects on its subcellular localization may not translate into significant modulation of its transcriptional activity.

## Discussion

MITF, TFEB, TFE3 and TFEC constitute the MiT-TFE subfamily of transcription factors featuring high structural homology. Their basic domains are identical and, unlike other members of the bHLHZip family, they share a three amino acid sequence in the HLHZip domain that enables them to restrict the formation of DNA-binding heterodimers to the MiT-TFE subfamily [22]. MITF, TFEB and TFE3 are expressed to some extent across melanoma tumors and cell lines, whereas TFEC is not [21,31]. Similarly, TFEB and TFE3 mRNA expression can be detected in normal melanocytes, albeit at lower levels than MITF [31,41]. Although the expression of TFEB is low at the mRNA level, there may be sufficient protein in the cells to have major effects, especially taking into consideration the biological role of TFEB and recent findings regarding low affinity vs high affinity binding sites across the genome for a given transcription factor [42]. Evidence for a degree of interplay between these factors comes from previous studies that have highlighted a role of the MiT-TFE factors in renal cell carcinoma. While the four MiT-TFE subfamily members are expressed at comparable levels in healthy human kidney, in subsets of renal cell carcinoma tumors the expression of these factors is significantly imbalanced. A fusion of *TFEB* located on chromosome 6 with the *Alpha* gene on

chromosome 11 resulting in an *AlphaTFEB* fusion gene, links *TFEB* with the regulatory regions upstream of the *Alpha* gene, leading to promoter substitution and a 60-fold increase in expression [43]. Additionally, translocations of the Xp11.2 region involving *TFE3* have been reported in up to 30–50% of pediatric papillary renal cell carcinoma cases [44]. The most commonly occurring translocations of this locus fuse the *TFE3* gene with the *ASPL* [45] or *PRCC* [46] genes, resulting in chimeric proteins with aberrant function. Interestingly, TFE3 depletion inhibited proliferation of a renal carcinoma cell line, whereas ectopic overexpression of MITF in the TFE3-depleted cells rescued proliferation [47]. Likewise, the proliferative defects induced by MITF knockout in an MITF-driven renal clear cell carcinoma model was restored by transfecting TFE3 [47], indicating that MITF and TFE3 may have partially redundant roles in regulating proliferation and survival. Redundancy between MITF and TFE3 was also suggested by the observation that loss of function mutation in either gene leads to normal bone development whereas simultaneous knockout of both factors resulted in the development of severe osteopetrosis in mice [48].

We show that overexpression of TFEB or MITF itself reduced the expression of endogenous MITF at both the mRNA and protein levels in two different melanoma cell lines (Fig 2 and S1 Fig). This indicates that MITF represses its own expression. This is consistent with ChIP-seq data showing that MITF binds to sequences within the *MITF* gene (Fig 1). MITF expression is regulated by multiple signaling mechanisms and transcription factors, including potential self-regulation. The peptide hormone α-MSH activates the MC1R receptor at the melanocyte membrane triggering cAMP signaling [49,50], which regulates the MITF-M promoter due to the cooperation of CREB with SOX10, a transcription factor that is expressed in several neural crest-derived cell lineages [51]. Various other transcription factors, such as beta-catenin, LEF1 and PAX3 have been reported to regulate MITF transcriptionally [52]. Previous studies using Northern blot analysis showed that α-MSH treatment resulted in increased *MITF* mRNA expression, peaking at two hours and entering a declining phase beyond this time point, indicating a homeostatic regulatory mechanism that was coupled with a decrease in protein expression beginning at 4 hours [50]. Of note, MITF has been shown to induce the expression of both HIF1-alpha and miR-148a, which in turn inhibit the expression of MITF, resulting in oscillatory levels of MITF expression that are required for an adequate physiological response to UVB exposure [53,54]. Whether MITF alone is sufficient for this negative feedback mechanism or if HIF1-alpha, miR-148a or other factors are also involved remains to be determined.

Our results show that MITF positively regulates the expression of TFEB. This is consistent with ChIP-seq data indicating that MITF binds intron 1 of *TFEB* [33]. Using reporter gene assays, we demonstrated that the increase in TFEB expression mediated by MITF is through direct binding to the CAGCTG motif located under the MITF ChIP-seq peak in intron 1 of *TFEB* (Fig 4). Our analysis of the MITF ChIP-seq dataset revealed that the CAGCTG 6-mer is as likely to be found under the peaks containing putative MITF binding sites as the canonical E-box and M-box, but more likely to be found than the CCCTTT motif used as a scramble control (Fig 4D). These data as a whole suggest that the non-canonical E-box CAGCTG is a DNA regulatory element that is bound by MITF and mediates the regulation of TFEB expression by MITF. Previous analysis of the MITF structure bound to the CACGTG and CATGTG sequences showed that Arg217 of MITF forms specific bonds with the two central bases of the E-box (CACGTG) motif, whereas it does not form base-specific bonds with the two central bases of the CATGTG M-box motif [22]. Instead, MITF forms specific bonds with the two bases flanking the 6-bp motif, -4 and +4, and with -3, -2 and +3, counting from the center of the motif, in the CATGTG M-box motif. This suggests that the two central bases within the subset of 6-bp E-box motifs are not always required for MITF binding, allowing a certain degree of flexibility for MITF's function as a transcription factor.

Recent studies have shown that the acetylation status of MITF impacts genomic occupancy as a means to modulate its transcriptional activity. Non-acetylated high DNA-binding-affinity MITF is able to bind a large pool of DNA loci including non-canonical degenerate motifs. In contrast, K243-acetylated MITF or the acetyl-mimetic K243Q mutant has low DNA-binding-affinity, yet robustly activates expression of melanocyte and melanoma target genes [42]. It is possible that acetylation of MITF affects binding to the non-canonical CAGCTG motifs found in TFEB. Furthermore, mTORC1 has been shown to phosphorylate and positively regulate the p300 acetyltransferase [55], which in turn acetylates MITF [42,50,56], suggesting that the mTOR pathway might be capable of modulating not only the subcellular localization of the MiT-TFE factors, but also shift their genomic occupancy towards high-affinity sites.

By immunostaining of cell preparations and confocal imaging we showed that in 501Mel cells TFEB is present in both the cytoplasm and the nucleus, whereas MITF-M is mostly nuclear, consistent with previous studies showing predominantly nuclear location of MITF-M [25]. Inhibition of mTOR promoted nuclear shuttling of TFEB. The effects of blocking mTOR activity on the subcellular location of TFEB have the potential to modulate the autophagy response, which has been linked to increased vesicle trafficking and chemoresistance in melanoma [57]. Consistent with this, in pancreatic ductal adenocarcinoma (PDA), but not in non-transformed human pancreatic ductal epithelial cells, the MiT-TFE factors have been shown to escape cytoplasmic retention mediated by mTOR regulation under fully fed conditions. This may enable PDA cells to constitutively induce autophagy activity, thus maintaining a high supply of amino acids [58]. Moreover, a subset of genes involved in endolysosomal trafficking and autophagy have been found to be overexpressed in melanoma [59], suggesting that in some tumor types and under certain conditions, high levels of autophagy activity can be beneficial for tumor survival and/or progression.

Altogether, our data is descriptive of a transcriptional cross-regulatory mechanism between MITF and TFEB that adds another layer of regulatory interactions beyond their ability to heterodimerize [14]. In addition, changes in the subcellular localization of TFEB, such as those mediated by mTOR, affect its transcriptional activity [19] and its regulation of *MITF* expression (Fig 6). The physiological relevance of their cross-regulation may be largely dependent on the relative abundance of each factor in a given tissue. Considering the prominent role of the MiT-TFE transcription factors in regulating various basic process as well as their roles in cancer, characterizing their expression patterns and how they are being modulated is instrumental for improving our understanding of their role in healthy tissue and in pathogenesis, and how best to identify targetable vulnerabilities in their action.

## Materials and methods

### Cell culture

Two human melanoma cell lines were used in this study, 501Mel cells (generously donated by Ruth Halaban) [60] and SkMel28 cells (#HTB-72, ATCC). HEK293T human embryonic kidney cells (#CRL-3216, ATCC) were used for transactivation assays. All cells were grown in DMEM medium (#10569–010, GIBCO) supplemented with 10% fetal bovine serum (FBS #10270–106, GIBCO). Cells were grown at 37˚C and 5% $CO_2$ and medium was changed two to three times per week.

### Plasmid constructs and cloning

In order to induce expression of the different transcription factors at will, we used an inducible piggyback (pBac) system. We generated inducible 501Mel cells by transfecting the cells with three pBac vectors, one containing GFP-tagged human MITF, TFEB, TFE3 or GFP alone, one

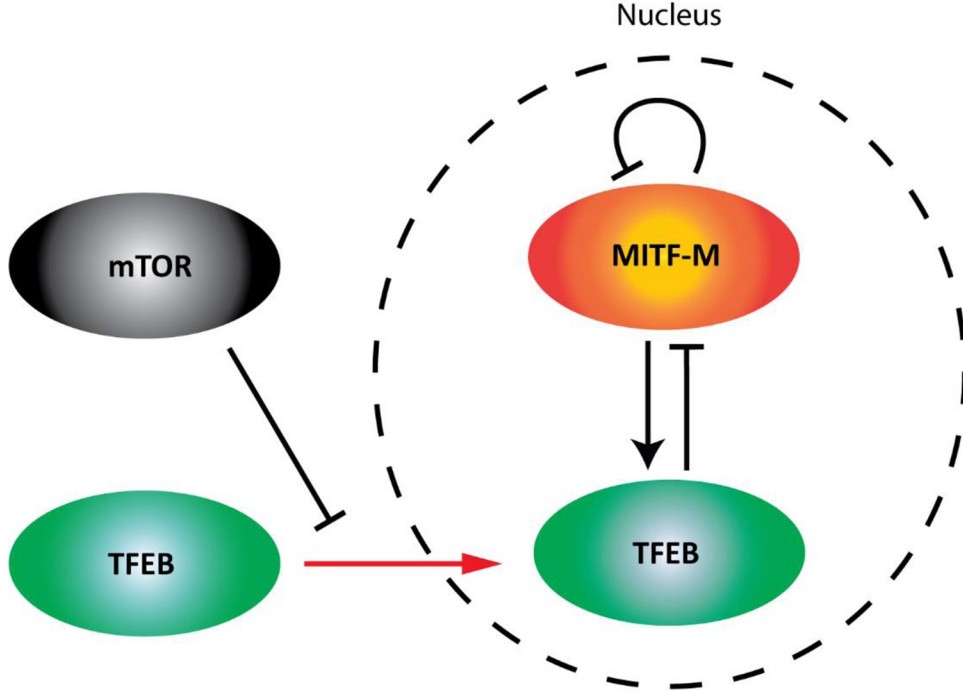

**Fig 6. Model of the cross-regulatory relationship between MITF and TFEB in melanoma.** MITF can inhibit its own gene expression and increase that of *TFEB*. TFEB can inhibit the gene expression of *MITF*. The nuclear localization of TFEB is increased upon mTOR inhibition, which enhances its inhibitory effects on *MITF* gene expression. In contrast, the MITF-M isoform expressed in melanoma cells is predominantly nuclear under basal conditions and its activity is less affected by mTOR regulation.

containing a reverse-tetracyclin transcription activator, and one containing transposase. The pBac vectors were a gift from Dr. Kazuhiro Murakami (Hokkaido University, Japan) [61]. The GFP-tagged MITF and TFEB cDNAs were amplified from plasmids pEGFP-N1-MITF-M (Addgene plasmid # 38131) and pEGFP-N1-TFEB (Addgene plasmid # 38119) [26], using the primers listed in S3 Table (pBac-EGFP), and then introduced into the pBac vector by restriction digestion using *Mlu* I and *Not* I sites and ligation at a 3:1 insert to backbone ratio using Instant Sticky-end Ligase Master Mix (M0370S, NEB). For the generation of the inducible 501Mel cells carrying the MITF-M-FLAG-HA pBac plasmid, a FLAG-tagged MITF-M cDNA was amplified from the p3XFLAG-CMV$^{TM}$-14 plasmid expressing mouse Mitf-M, kindly provided by Colin Goding (Ludwig Institute, Oxford University, UK), using the primers listed in S3 Table (pBac-MITF-M-FLAG-HA), and then introduced into the pBac vector by restriction digestion with *EcoR* I and *Spe* I. Cells transfected with the pBac plasmids were cultured in DMEM supplemented with 10% FBS and kept under G418 selection (#10131–035, GIBCO) for 8 days to obtain stable cell lines that are inducible by adding 0.2 μg/mL doxycycline to the culture medium. The p3XFLAG-CMV$^{TM}$-14 construct expressing mouse Mitf-M (MITF-M-FLAG) was the one used for MITF-M overexpression in the transactivation assays. The R214-217A mutation was introduced into MITF using the Q5 Site-directed Mutagenesis Kit (New England Biolabs, Ipswich, MA) according to the manufacturer's instructions. The pGL3-Basic vector (#E1751, Promega) was used as a control for *Tyrosinase* (pTYR) transactivation, whereas pGL3-Promoter vector (#E1761, Promega) was used as a control for the *TFEB* intron 1 enhancer (TFEB-int1) and mutated *TFEB* intron 1 (TFEB-int1-mut) promoter constructs. The *TYR* promoter in a luciferase reporter plasmid was constructed using a 380 bp region at

the promoter of the human *TYR* gene (bases -382 and -3 upstream of the TSS at location +1). This sequence was then cloned into a pGL3-Basic vector upstream of the luciferase reporter gene and verified by Sanger sequencing. The fragment containing TFEB-int1 was amplified from human genomic DNA using the primers listed in S3 Table. The 853 bp fragment was blunt-ligated into Nhe1-site of the pGL3-Promoter vector. TFEB-int1-mut was generated from the TFEB-int1 luciferase reporter plasmid by mutating CAGCTGA to CCCTTTA using *in vitro* mutagenesis. All the constructs were confirmed by sequencing (Genewiz, Essex, UK). All the mutagenesis primers are listed in S3 Table.

## MITF knockdown cell lines

In order to be able to induce knockdowns of MITF at will, we generated two piggybac (pBac) constructs containing miRNAs under the regulation of an inducible promoter. Skmel28 doxycycline-inducible MITF knockdown cell lines were generated using a piggybac transposable vector pPBhCMV_1-miR(BsgI)-pA-3 obtained from Dr. Kazuhiro Murakami (Hokkaido University) [61]. MicroRNAs target sequences were selected using the BLOCK-iT RNAi Designer, specifically targeting both MITF-M exon 2 (miR(MITF-X2) and 8 (miR(MITF-X8). The non-targeting control miR-NTC was used as a negative control. The BLOCK-iT RNAi Designer was also used to design primers for inserting the pre-miRNA (including a mature miRNAi sequence, terminal loop and incomplete sense targeting sequence required for the formation of the stem-loop structure) RNAi into the murine miR-155 cassette in the pBac vector pPBhCMV_1-miR(BsgI)-pA-3 containing the reverse tetracycline transcription activator. Sequences of the mature miRNAs and the primers used for the generation of the pre-miRNAs are listed in S4 Table. Primers were annealed by initial denaturation at 95°C followed by slow cooling in a water bath forming a short double stranded DNA with overhangs matching a BsgI overhang. The backbone vector was digested with BsgI (#R05559S, NEB) and the vector DNA purified after running the DNA on an agarose gel. The backbone vector and the annealed primers were ligated at 15:1 insert to backbone molar ratio using Instant Sticky-end Ligase Master Mix (M0370S, NEB). The ligation products were transformed into high-competent cells and the plasmid DNA isolated from the individual clones were screened as described above in Mutagenesis and cloning. For generation of miR-MITF cell lines, Skmel28 cells were transfected with the transposase-containing plasmid pA-CAG-pBase, the plasmids pPBhCMV_1-miR(MITF_X2)-pA and pPBhCMV_1-miR (MITF_X8)-pA encoding miRNA targeting two different exons of MITF and the plasmid pPB-CAG-rtTA-IRES-Neo (10:5:5:1) that confers resistance to neomycin. For miR-NTC cell lines, Skmel28 cells were transfected with pA-CAG-pBase, pPBhCMV_1-miR(NTC)-pA encoding a non-targeting miRNA and pPB-CAG-rtTA-IRES-Neo (10:10:1). After 48 hours of transfection, miR-MITF, miR-NTC vector and non-transfected cells were selected with 0.5mg/ml G418 (#10131–035, GIBCO) for 2 weeks and 1μg/ml of doxycycline was used for induction.

## Transfection of plasmids

Cells were cultured in 6 or 12-wells plates one day before transfecting them with 2 μg of plasmid DNA and 6 μL transfection reagent FuGENE HD (#E2311, Promega) in 100 μL of serum-free culture medium per mL of cell culture medium. The medium with the transfection complexes was removed after 24 hours and replaced with fresh culture medium. 48 hours after transfection, cells were harvested for RNA or protein extraction.

## RNAi treatment

Cells were cultured in 6 or 12-wells plates one day prior to transfection with the appropriate siRNA pools. Cells were transfected with 10 nM siRNA pools and 1 μL Lipofectamine

RNAiMAX (#13778075, Invitrogen) transfection reagent in 100 μL of Optimem Pro (#31985–062, GIBCO) transfection medium per mL of cell culture medium. Cells were harvested for RNA or protein extraction 2 days after transfection. The siRNAs used for the procedure were the following: siRNA pool for human MITF (#4390824, ID S8792, Ambion), siRNA pool for human TFEB (#M-009798-02, Dharmacon) and a control siRNA (#4390843, Ambion).

## Protein extraction and immunoblotting

For total protein extraction, cells were cultured in 6 or 12-wells plates and lysed in Laemmli sample buffer and boiled at 95˚C for 5 minutes. The samples were then run on 8% or 10% gels and blotted onto a 0.2 μm PVDF membrane (#88520, Thermo Scientific). The membranes were blocked with 3% BSA in TBS-T (0.1% Tween 20 in TBS) for 1 hour at room temperature, and stained overnight at 4˚C with 3% BSA in TBS-T and one of the following primary antibodies: MITF (MS771-PABX, Thermo Scientific), TFEB (#4240, CST) (#2775, CST), GFP (#ab290, Abcam), FLAG (#F3165, Sigma Aldrich) and Actin (MAB1501, Millipore). Membranes were washed with TBS-T and stained for 1 hour at room temperature with fluorescent secondary antibodies: anti-mouse IgG(H+L) DyLight 800 conjugate (#5257, CST) and anti-rabbit IgG(H+L) DyLight 680 conjugate (#5366, CST). The images were captured using Odyssey CLx Imager (LI-COR Biosciences).

## Real-time quantitative PCR for gene expression analysis

TRIzol reagent (#15596–026, Ambion) followed by isopropanol precipitation was used for total RNA extraction. The cDNA was generated according to the manufacturer's instructions with High-Capacity cDNA Reverse Transcription Kit (#4368814, Applied Biosystems). Primers for RT-qPCR were designed using NCBI Primer BLAST (S2 Table), and RT-qPCR performed with SensiFAST SYBR Lo-ROX Kit (#BIO-94020, Bioline) using a CFX384 Touch Real-Time PCR Detection System (Bio-Rad). The RT-qPCR reactions were performed using 1 ng/μL cDNA per reaction in technical triplicates and the fold change in gene expression was calculated with the 2(-Delta Delta C(T)) method [62], normalized to the geometrical mean of Actin and human ribosomal protein lateral stalk subunit P0 (RPLP0) expression. Standard curves for primer efficiency were calculated using the formula $E = 10^{(-1/slope)}$ for each primer pair.

## Immunostaining and confocal imaging

501Mel and Skmel28 cells ($3x10^4$ per well) were cultured for 48 hours in 8-well chamber slides (#354108 from Falcon). For the inducible 501Mel cell lines, 0.2 μg/mL of doxycycline were added to the cell culture medium. At day 2, cells were fixed for 2 min with 2% paraformaldehyde (PFA) in cell culture medium and then for 15 min with 4% PFA in PBS. For imaging of the overexpressed GFP-tagged factors, cells were washed 3 times with PBS, followed by DAPI staining (#D-1306, Life Technologies) and two additional washes with PBS. The chambers were then removed and the samples allowed to dry prior to mounting in Fluoromount-G™ (#00-4958-02, Invitrogen). For immunostaining of the endogenous MiT/TFE factors, following the treatment with the Torin-1 (#4247, Tocris) inhibitor, cells were washed once with PBS after fixation, then permeabilized for 8 min with 0.1% Triton X-100 in PBS, followed by three washes with PBS. They were then blocked with blocking buffer (5% normal goat serum, 0.05% Triton X-100 and 0.25% BSA in PBS) for 1 hour at room temperature, and stained overnight at 4˚C with 0.25% BSA in PBS antibody buffer containing the primary antibodies: MITF (MS771-PABX, Thermo Scientific) and TFEB (#4240, CST) (#2775, CST). The cells were then washed three times with PBT (0.1% Tween-20 in PBS) and stained for 1 hour at room

temperature with the Alexa Fluor 546 goat anti-mouse IgG(H+L) (#A11003, Invitrogen) or the Alexa Fluor 488 goat anti-rabbit IgG(H+L) (#A11070, Invitrogen) fluorescent secondary antibodies diluted in PBT. Subsequently, cells were washed twice with PBT and once with PBS and finally stained with DAPI and prepared for imaging as previously described. Imaging was performed using a FluoView FV1200 laser scanning confocal microscope (Olympus) equipped with a PlanApo N 60X/1.40 ∞/0.17 Oil Objective.

## Transcription activation assays

HEK293T cells ($1.5x10^4$ per well) were seeded in white 96-well plates (#781965, BRAND) and cultured for 24h prior to transfection (FuGENE, Promega) with 33 ng of a construct carrying the relevant regulatory region (Tyr, TFEB-int1 or TFEB-Int1-mut) coupled to the luciferase reporter, 33 ng of an MITF-M construct and 33 ng of a pRL Renilla control reporter vector. Cells were assayed 24 hours after transfection using the Luciferase DualGlo kit (E2940, Promega) as described by the manufacturer. The luminescence activity was measured in a multimode microplate reader (GloMax, Promega) with a 300-millisecond reading per well. The luciferase signal of each sample was normalized to the renilla signal for transfection efficiency and cell viability. The pGL3-Basic (#E1751, Promega) or pGL3-Promoter (#E1761, Promega) vectors were used in order to calculate the fold induction of each respective regulatory element activity. Three technical replicates per sample were included and the assay was performed in at least three biological replicates. Error bars indicate SEM and statistical significance was assessed with student's t-tests.

## Electrophoretic Mobility Shift Assay (EMSA)

Two DNA fragments were generated for the EMSA studies by synthesizing the oligos E-box-Fw (containing the sequence 5'-AAA GTC AGT CAC GTG CTT TTC AGA-3') and E-box-Rv (5'-GTC TGA AAA GCA CGT GAC TGA CTT T-3') for the canonical E-box (containing the CACGTG motif). CAGCTG-box-Fw (containing the sequence 5'-AAA GTC AGT CAG CTG ATT TTC AGA-3') and CAGCTG-box-Rv (5'-GTC TGA AAA TCA GCT GAC TGA CTT T-3') were synthesized for the CAGCTG motif-containing probe. Subsequently, the oligos were allowed to anneal and labeled with α-$^{32}$P-dCTP, (#BLU013H100UC, PerkinElmer). The labeled probes were purified on Sephadex G-25 Quick Spin columns (#11273922001, Roche).

The EMSA was performed according to Pogenberg et al. [22]. Briefly, the MITF protein was expressed from a plasmid containing wild type MITF-M under a T7 promoter, using the TNT-T7 Quick Coupled Transcription/Translation System (#L1170, Promega). 2 ul of TNT-translated MITF was preincubated in a buffer containing 20 ng of cold probe poly(dI–dC), 10% fetal calf serum, 2 mM MgCl2, and 2 mM spermidine for 15 min on ice. For supershift assays, 0.5 mg of mouse monoclonal antiMITF (C5) antibody (#ab12039, Abcam) were added and incubated on ice for 30 min. Then, 50,000 counts per minute (cpm) of each of the two $^{32}$P-labeled probes in a binding buffer containing 10 mM Tris (pH 7.5), 100 mM NaCl, 2 mM dithiothreitol, 1 mM EDTA, 4% glycerol, and 80 ng/mL salmon sperm DNA were added to the preincubated MITF protein solution in a total reaction volume of 20 ul and incubated for 10 min at room temperature. The resulting DNA–protein complexes were resolved on 4.2% non-denaturing polyacrylamide gels, placed on a storage phosphor screen, and then scanned using a Typhoon PhosphorImager 8610 (Molecular Dynamics).

## ChIP-seq data analysis and motif scan

Raw FASTQ files for MITF ChIP-seq were retrieved from GEO archive under the accession number GSE50681 and GSE61965 and subsequently mapped to the human reference genome

hg19 using bowtie 2 [63]. Then the aligned reads from bowtie2 were used to call peaks with MACS [64]. Following, wig and bed files were generated using the p-value<0.05. For visualization of the peaks, wig files were uploaded on the IGV genome browser [65]. For scanning the motifs, DNA sequences corresponding to MITF peaks were extracted from the UCSC genome browser. The R package Biostrings [66] was used to scan each motif and the resulting number of motifs for each peak sequence were plotted in R.

## Statistical analysis

Results are represented as the mean from three or more independent experiments with standard error of the mean (SEM). Graphpad Prism 7 was used for all the statistical analyses. Analyses of RT-qPCR and Western blot upon overexpression of the pBac factors were performed using multiple t-tests comparing each cell mean to the control cell mean and multiple comparison correction by Holm-Sidak method with statistical significance set as *P<0.05. Analysis of the transactivation assays was performed using one-way analysis of variance (ANOVA). All the remaining statistical analyses were performed using two-way ANOVA. ANOVA analyses were performed comparing each cell mean to the control cell mean and multiple comparison correction by Dunnett method for one-way ANOVA and correction by Sidak method for two-way ANOVA with statistical significance set as *P<0.05.

## Supporting information

**S1 Fig. MITF and TFEB modulate each other's expression upon overexpression in 501Mel and Skmel28 cells.** (A) RNA expression of *MITF* and *TFEB* after overexpression of GFP-tagged MITF (+) in 501Mel cells compared to empty vector control. (B) RNA expression of *MITF* and *TFEB* after overexpression of GFP-tagged MITF (+) or TFEB in Skmel28 cells compared to empty vector control.
(PDF)

**S2 Fig. siRNA-mediated knockdown of MITF modulates endogenous *TFEB* expression in Skmel28 cells.** The expression of *MITF* and *TFEB* as determined by RT-qPCR after siRNA knockdown of each factor, compared to control siRNA in Skmel28 cells; two independent experiments are shown. Bars represent SEM. * indicates significance at p<0.05.
(PDF)

**S3 Fig. R214-217A MITF fails to transactivate *TYR* or *TFEB* regulatory elements.** HEK293T cells were transiently co-transfected with a p3XFLAG-CMV-14 construct with or without R214-217A MITF-M (empty vector, EV) and luciferase constructs and assayed for luciferase activity after 24h. Luminescence signal is expressed as fold change over an empty reporter for *Tyrosinase* and a reporter containing the element from *TFEB* intron 1 in front of a minimal SV40 promoter followed by luciferase. Error bars represent the SEM of three experiments. * indicates significance at p<0.05.
(PDF)

**S4 Fig. mTOR signaling affects the subcellular localization of MITF in Skmel28 cells.** Immunofluorescence images of human Skmel28 cells after treatment with vehicle (DMSO) or an mTOR inhibitor (Torin-1, 1 μM, 3 hours), showing endogenous TFEB and MITF proteins in green.
(PDF)

**S1 Table. MITF, TFEB and TFE3 but not TFEC are expressed in 501Mel and Skmel28 cells.**
(PDF)

**S2 Table. Gene-specific primers used for RT-qPCR.**
(PDF)

**S3 Table. Primers used for cloning and mutagenesis of plasmid constructs.**
(PDF)

**S4 Table. miRNA sequences and primers used for the MITF knockdown cell lines.**
(PDF)

**S1 File.**
(PDF)

## Acknowledgments

We thank Colin Goding for critical comments on the manuscript.

## Author Contributions

**Conceptualization:** Josué Ballesteros-Álvarez, Margrét Helga Ögmundsdóttir.

**Formal analysis:** Josué Ballesteros-Álvarez, Ramile Dilshat.

**Funding acquisition:** Eiríkur Steingrímsson.

**Investigation:** Josué Ballesteros-Álvarez, Ramile Dilshat, Valerie Fock, Katrín Möller, Ludwig Karl.

**Methodology:** Josué Ballesteros-Álvarez, Ramile Dilshat, Valerie Fock.

**Resources:** Lionel Larue.

**Supervision:** Eiríkur Steingrímsson.

**Visualization:** Josué Ballesteros-Álvarez.

**Writing – original draft:** Josué Ballesteros-Álvarez, Eiríkur Steingrímsson.

**Writing – review & editing:** Josué Ballesteros-Álvarez, Eiríkur Steingrímsson.

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
