## [Decision Letter · Decision Letter 0]

19 Jun 2020

PONE-D-20-15262

MITF and TFEB cross-regulation in melanoma cells

PLOS ONE

Dear Dr. Steingrimsson,

Thank you for submitting your manuscript to PLOS ONE. After careful consideration, we feel that it has merit but does not fully meet PLOS ONE’s publication criteria as it currently stands. Therefore, we invite you to submit a revised version of the manuscript that addresses the points raised during the review process.

Please respond to all critique, point-by-point. In particular:

- Consult databases that provide insight into the expression of the MiT-TFE gene family.

- Discuss pool of low-affinity bdg. sites of MITF and the possible influence of acetylation.

We look forward to receiving your revised manuscript.

Kind regards,

Klaus Roemer

Academic Editor

PLOS ONE

Journal Requirements:

2. PLOS ONE now requires that authors provide the original uncropped and unadjusted images underlying all blot or gel results reported in a submission’s figures or Supporting Information files. This policy and the journal’s other requirements for blot/gel reporting and figure preparation are described in detail at https://journals.plos.org/plosone/s/figures#loc-blot-and-gel-reporting-requirements and https://journals.plos.org/plosone/s/figures#loc-preparing-figures-from-image-files.

When you submit your revised manuscript, please ensure that your figures adhere fully to these guidelines and provide the original underlying images for all blot or gel data reported in your submission.

See the following link for instructions on providing the original image data: https://journals.plos.org/plosone/s/figures#loc-original-images-for-blots-and-gels

Reviewers' comments:

Reviewer's Responses to Questions

**Comments to the Author**

1. Is the manuscript technically sound, and do the data support the conclusions?

Reviewer #1: Yes

Reviewer #2: Yes

2. Has the statistical analysis been performed appropriately and rigorously? 

Reviewer #1: Yes

Reviewer #2: Yes

3. Have the authors made all data underlying the findings in their manuscript fully available?

Reviewer #1: Yes

Reviewer #2: Yes

4. Is the manuscript presented in an intelligible fashion and written in standard English?

Reviewer #1: Yes

Reviewer #2: Yes

5. Review Comments to the Author

Reviewer #1: The manuscript is interesting and is well structured in general. The Authors conducted a number of assays using different techniques to reveal MITF and TFEB cross-regulation in melanoma cells. Moreover, in my opinion it could be interesting for a reasonable number of scientists since the studied cross-regulatory interactions need to be taken into account when considering pathways regulated by these proteins - e.g. as a potential mechanism underlying antimelnoma effect of the new treatment options. I recommend publication of this study in its present form.

Reviewer #2: This manuscript sets out to investigate the potential for cross regulation of the MiT-TFE family of genes, specifically how the melanocyte master regulator MITF may be influenced by the other family members. Previous studies have shown that the MITF mRNA is expressed at much higher levels than the TFE3, TFEB and TFEC genes in human melanoma cell lines and tumors. The authors have chosen to look at two melanoma cell lines 501Mel and Skmel28 to conduct a series of experiments, and first exclude the potential of TFEC for regulation as its expression is undetectable. Through the examination of ChIP-seq data it was found that MITF can bind the MITF and TFEB loci regulatory regions but less likely interact with TFE3. From these initial experiments the focus was then on the cross regulation of MITF and TFEB.

The interaction of these genes at both the mRNA and protein level was studied by transient overexpression of each protein in these melanoma lines. This found that the MITF protein induced TFEB but reduced the endogenous MITF transcript, with TFEB reducing MITF. The effect of TFEB on endogenous TFEB is not addressed? These results were validated using an siRNA KD approach. This negative feedback loop is illustrated in Figure 6. The binding of MITF to the non-consensus E-box CAGCTG sequence within the region of the first intron of TFEB was then demonstrated using a luciferase reporter construct and by DNA binding EMSA assay. This sequence element was then further analysed in the ChIP-seq dataset for MITF confirming it as an efficient binding site. The nuclear/cytoplasmic localisation of MITF and TFEB and effect of mTOR were examined by immunostaining and Torin-1 treatment of melanoma cells. This data clearly demonstrates MITF as a major presence in the nuclear fraction, whereas TFEB located in the cytoplasm and nucleus which the can move to the nucleus following inhibition of mTOR by Torin-1.

Things to consider:

1. In the final sentence of the manuscript “… improving our understanding of their role in healthy tissue …” does not consider what the expression status of the MiT-TFE family may be in normal melanocytic cells. The authors may consider if there are any databases examining expression of these genes in different tissues could be consulted e.g.

Melanocytes in the skin--comparative whole transcriptome analysis of main skin cell types.

Reemann P, Reimann E, Ilmjärv S, Porosaar O, Silm H, Jaks V, Vasar E, Kingo K, Kõks S.PLoS One. 2014 Dec 29;9(12):e115717.

This data was extracted from.

S1 Table. RPKM values of genes we detected in MC, KC, FB and the whole skin.

Melanocytes

genes MC_1 MC_2 MC_3 MC_4

ENSG00000187098 199.197 156.316 191.509 167.668 MITF

ENSG00000112561 0.400417 1.12329 0.798733 0.35943 TFEB

ENSG00000068323 1.13973 1.78558 1.92091 1.13326 TFE3

ENSG00000105967 0.0451891 0.048543 0.0621599 0.0607503 TFEC

2. The recent paper describing acetylation of MITF, in which the senior author of this manuscript is a co-author, should at least be cited. Some discussion of how MITF has a pool of low affinity binding sites that may keep it in the nucleus and how may this differ or be similar for the other MiT-TFE family, and effects of acetylation on other family members such as TFEB could be included.

Tuning Transcription Factor Availability through Acetylation-Mediated Genomic Redistribution.

Louphrasitthiphol P, Siddaway R, Loffreda A, Pogenberg V, Friedrichsen H, Schepsky A, Zeng Z, Lu M, Strub T, Freter R, Lisle R, Suer E, Thomas B, Schuster-Böckler B, Filippakopoulos P, Middleton M, Lu X, Patton EE, Davidson I, Lambert JP, Wilmanns M, Steingrímsson E, Mazza D, Goding CR.Mol Cell. 2020 Jun 4:S1097-2765(20)30345-2.

6. PLOS authors have the option to publish the peer review history of their article (what does this mean?). If published, this will include your full peer review and any attached files.

Reviewer #1: No

Reviewer #2: No

---

## [Author Response · Author response to Decision Letter 0]

6 Jul 2020

Dear editor,

We thank PLOS One for the opportunity to resubmit our manuscript after addressing the concerns of reviewers and editor. We also thank the reviewers for their critical comments and analysis. Following are the specifics of the changes we made:

1. We have made sure that the manuscript fits the PLOS ONE style requirements and renamed the files appropriately.

2. The uncropped and unadjusted images are provided for all images as supporting information.

3. The data previously referred to as “data not shown” is now available on Figshare (DOI: 10.6084/m9.figshare.12568646).

Reviewer #2:

Comment 1: In the final sentence of the manuscript “… improving our understanding of their role in healthy tissue …” does not consider what the expression status of the MiT-TFE family may be in normal melanocytic cells. The authors may consider if there are any databases examining expression of these genes in different tissues could be consulted e.g. Melanocytes in the skin--comparative whole transcriptome analysis of main skin cell types. Reemann P, Reimann E, Ilmjärv S, Porosaar O, Silm H, Jaks V, Vasar E, Kingo K, Kõks S.PLoS One. 2014 Dec 29;9(12):e115717. This data was extracted from.

S1 Table. RPKM values of genes we detected in MC, KC, FB and the whole skin.

Melanocytes

genes MC_1 MC_2 MC_3 MC_4

ENSG00000187098 199.197 156.316 191.509 167.668 MITF

ENSG00000112561 0.400417 1.12329 0.798733 0.35943 TFEB

ENSG00000068323 1.13973 1.78558 1.92091 1.13326 TFE3

ENSG00000105967 0.0451891 0.048543 0.0621599 0.0607503 TFEC

We have consulted our own array-based gene expression data where the expression of MITF, TFEB and TFE3 was determined in melanocytes and melanoma cells. We have introduced the following into the results section, page 6, lines 72-76:

“Furthermore, analysis of gene expression in 23 human melanoma cell lines as well as in normal human epidermal melanocytes (NHEM) using a microarray platform revealed that the expression of TFEB and TFE3 was roughly 50-fold lower than that of MITF, whereas expression of TFEC mRNA was about 850-fold lower than that of MITF [21,31].”

We also added the following in the Discussion section, page 17, lines 317-317:

“MITF, TFEB and TFE3 are expressed to some extent across melanoma tumors and cell lines, whereas TFEC is not [21,31]. Similarly, TFEB and TFE3 mRNA expression can be detected in normal melanocytes, albeit at lower levels than MITF [31,41].”

Comment 2. The recent paper describing acetylation of MITF, in which the senior author of this manuscript is a co-author, should at least be cited. Some discussion of how MITF has a pool of low affinity binding sites that may keep it in the nucleus and how may this differ or be similar for the other MiT-TFE family, and effects of acetylation on other family members such as TFEB could be included.

Tuning Transcription Factor Availability through Acetylation-Mediated Genomic Redistribution.

Louphrasitthiphol P, Siddaway R, Loffreda A, Pogenberg V, Friedrichsen H, Schepsky A, Zeng Z, Lu M, Strub T, Freter R, Lisle R, Suer E, Thomas B, Schuster-Böckler B, Filippakopoulos P, Middleton M, Lu X, Patton EE, Davidson I, Lambert JP, Wilmanns M, Steingrímsson E, Mazza D, Goding CR.Mol Cell. 2020 Jun 4:S1097-2765(20)30345-2.

We have entered a paragraph discussing this interesting study in the page 20, between lines 374-383, of the Discussion that reads the following: 

“Recent studies have shown that the acetylation status of MITF impacts genomic occupancy as a means to modulate its transcriptional activity. Non-acetylated high DNA-binding-affinity MITF is able to bind a large pool of DNA loci including non-canonical degenerate motifs. In contrast, K243-acetylated MITF or the acetyl-mimetic K243Q mutant has low DNA-binding-affinity, yet robustly activates expression of melanocyte and melanoma target genes (54). It is possible that acetylation of MITF affects binding to the non-canonical CAGCTG motifs found in TFEB. Furthermore, mTORC1 has been shown to phosphorylate and positively regulate the p300 acetyltransferase (55), which in turn acetylates MITF (49, 54, 56), suggesting that the mTOR pathway might be capable of modulating not only the subcellular localization of the MiT-TFE factors, but also shift their genomic occupancy towards high-affinity sites.”

In addition, we reason that this study may be relevant to the observation that although the expression of TFEB is low in melanocytes and in some melanoma tumor and cell lines, the protein levels of this factor may be sufficient for binding and activation for a subet of high-affinity binding sites. This statement can be found in the Discussion section, pages 17-18 between lines 317-320:

“Although the expression of TFEB is low at the mRNA level, there may be sufficient protein in the cells to have major effects, especially taking into consideration the biological role of TFEB and recent findings regarding low affinity vs high affinity binding sites across the genome for a given transcription factor [42].“

Best regards

---

## [Editor Report · Decision Letter 1]

19 Aug 2020

MITF and TFEB cross-regulation in melanoma cells

PONE-D-20-15262R1

Dear Dr. Steingrimsson,

We’re pleased to inform you that your manuscript has been judged scientifically suitable for publication and will be formally accepted for publication once it meets all outstanding technical requirements.

Kind regards,

Klaus Roemer

Academic Editor

PLOS ONE
---

## [Editor Report · Acceptance letter]

21 Aug 2020

PONE-D-20-15262R1 

MITF and TFEB cross-regulation in melanoma cells 

Dear Dr. Steingrímsson:

I'm pleased to inform you that your manuscript has been deemed suitable for publication in PLOS ONE. Congratulations! Your manuscript is now with our production department. 

Kind regards, 

on behalf of

Dr. Klaus Roemer 

Academic Editor

PLOS ONE